# Vitamin D and the Ability to Produce 1,25(OH)_2_D Are Critical for Protection from Viral Infection of the Lungs

**DOI:** 10.3390/nu14153061

**Published:** 2022-07-26

**Authors:** Juhi Arora, Devanshi R. Patel, McKayla J. Nicol, Cassandra J. Field, Katherine H. Restori, Jinpeng Wang, Nicole E. Froelich, Bhuvana Katkere, Josey A. Terwilliger, Veronika Weaver, Erin Luley, Kathleen Kelly, Girish S. Kirimanjeswara, Troy C. Sutton, Margherita T. Cantorna

**Affiliations:** 1Department of Veterinary and Biomedical Sciences, The Pennsylvania State University, University Park, PA 16802, USA; jua268@psu.edu (J.A.); drp5323@psu.edu (D.R.P.); mjn5181@psu.edu (M.J.N.); cif5202@psu.edu (C.J.F.); khr114@psu.edu (K.H.R.); jinpeng.wong1990@gmail.com (J.W.); nef5148@psu.edu (N.E.F.); bxk33@psu.edu (B.K.); josey.terwilliger@gmail.com (J.A.T.); vcw100@psu.edu (V.W.); gsk125@psu.edu (G.S.K.); 2Animal Diagnostic Laboratory, The Pennsylvania State University, University Park, PA 16802, USA; ehl5008@psu.edu (E.L.); kmk6898@psu.edu (K.K.)

**Keywords:** vitamin D, influenza, SARS-CoV-2, lung, inflammation

## Abstract

Vitamin D supplementation is linked to improved outcomes from respiratory virus infection, and the COVID-19 pandemic renewed interest in understanding the potential role of vitamin D in protecting the lung from viral infections. Therefore, we evaluated the role of vitamin D using animal models of pandemic H1N1 influenza and severe acute respiratory syndrome coronavirus-2 (SARS-CoV-2) infection. In mice, dietary-induced vitamin D deficiency resulted in lung inflammation that was present prior to infection. Vitamin D sufficient (D+) and deficient (D−) wildtype (WT) and D+ and D− Cyp27B1 (Cyp) knockout (KO, cannot produce 1,25(OH)_2_D) mice were infected with pandemic H1N1. D− WT, D+ Cyp KO, and D− Cyp KO mice all exhibited significantly reduced survival compared to D+ WT mice. Importantly, survival was not the result of reduced viral replication, as influenza M gene expression in the lungs was similar for all animals. Based on these findings, additional experiments were performed using the mouse and hamster models of SARS-CoV-2 infection. In these studies, high dose vitamin D supplementation reduced lung inflammation in mice but not hamsters. A trend to faster weight recovery was observed in 1,25(OH)_2_D treated mice that survived SARS-CoV-2 infection. There was no effect of vitamin D on SARS-CoV-2 N gene expression in the lung of either mice or hamsters. Therefore, vitamin D deficiency enhanced disease severity, while vitamin D sufficiency/supplementation reduced inflammation following infections with H1N1 influenza and SARS-CoV-2.

## 1. Introduction

Low vitamin D status is associated with poorer outcomes following acute respiratory diseases including influenza [1]. Vitamin D supplements are touted as being useful in high doses for reducing the severity of seasonal influenza [2,3,4]. The recent emergence of severe acute respiratory syndrome (SARS)-coronavirus (CoV)-2 and the ongoing pandemic led to a renewed interest in high-dose vitamin D supplements to prevent and treat severe SARS-CoV-2 disease (i.e., COVID-19) [4]. Infection with SARS-CoV-2 results in local and systemic inflammation that when controlled may enhance survival and clinical outcomes of COVID-19 [5]. Severe respiratory illness can also be caused by influenza viruses or co-infection with influenza and coronaviruses. An association between low vitamin D status and severe COVID-19 was postulated, and accordingly, vitamin D supplementation was proposed to be beneficial to treat COVID-19 [6]. A recent systemic review concluded that low circulating levels of vitamin D (serum 25(OH)D, 25D) were associated with more severe symptoms and higher mortality in patients with COVID-19 [7]. Interventions that control the lung inflammatory response to viruses have the potential to benefit the global population.

Vitamin D, and the active form of vitamin D (1,25(OH)_2_D, 1,25D), has been implicated to play a role in the anti-viral response; however, this effect may be specific to different viruses. For example, 1,25D treatment of T cells from human immunodeficiency virus (HIV)-infected individuals in vitro resulted in a decrease in viral RNA transcription by a direct reduction in NF-κB, which reactivates proviral HIV [8]. Conversely, 1,25D treatment of respiratory syncytial virus (RSV)-infected human epithelial cells in vitro did not affect viral replication [9]. Production of antimicrobial peptides such as cathelicidin, β-defensin etc. and production of cytokines such as TNFα, IL-5, IL-1β, IL-6, IL-10, and type I interferons at the mucosal surface are important parts of the innate immune response against viruses [10,11]. Several studies show that 1,25D and other vitamin D analogs induce cathelicidin production in response to virus infection [12]. Cathelicidin LL-37 was shown to bind and kill viruses including influenza viruses in vitro [13,14,15,16,17]. Therefore, it is possible that vitamin D through the induction of cathelicidin could directly target SARS-CoV-2 and influenza. 1,25D also limits inflammatory responses by decreasing IFNγ, IL-6, and TNFα: however, the effects of 1,25D to reduce inflammation could be detrimental for the ability of the host to clear some viruses [18,19]. 

Lung epithelial cells are vitamin D targets since they express the vitamin D receptor (VDR) and are regulated by 1,25D treatments. In mice, 1,25D treatments reduced inflammation following lipopolysaccharide-induced lung injury and regulated angiotensin converting enzyme (ACE) expression in the rat lung epithelium [20]. In mice, infection with an influenza H9N2 virus induced mRNA for the VDR in the lung and 1,25D-treated animals had reduced lung inflammation [21]. Treating mice with the 1,25D precursor, 25hydroxyvitamin D (25D), also protected mice from subsequent H1N1 influenza infection [22]. 1,25D treatment of human bronchial epithelial cells suppressed IL-6 and protected the cells from oxidative damage [23]. VDR knockout (KO) mice had reduced expression of tight junction proteins such as zonula occludens-1, occludin, and claudins (2,4, and 12), suggesting an important role for the VDR in maintaining the integrity of the lung [24]. Vitamin D has direct effects on the lung epithelium, and 1,25D suppresses inflammation in the lung.

Extensive association studies in humans led to the proposal that high dose vitamin D supplements could be beneficial for protection from severe influenza and SARS-CoV-2 infections. Since cause and effect are extremely difficult to determine in human studies, we sought to evaluate the effects of vitamin D on the lung anti-viral response in animal models. The data from mice and hamsters suggest that vitamin D supplementation reduces inflammation in the lung following pandemic H1N1 and SARS-CoV-2 infection. D− mice had lung inflammation even without infection. We showed previously that feeding D− diets to mice that cannot produce 1,25D (Cyp KO) resulted in severe vitamin D deficiency [25]. Influenza disease was greatest in D− Cyp KO, and the least amount of disease was in D+ WT mice. The survival of D+ Cyp KO mice was significantly less than D+ WT mice following an influenza infection. Vitamin D supplementation reduced lung inflammation and *Ifnb* expression in the lung of mice following SARS-CoV-2 infection. Vitamin D treatments had no effect on the expression of viral RNA for either SARS-CoV-2 or H1N1 influenza in the lungs of hamsters or mice. Instead, the data support an important role for vitamin D and 1,25D in controlling the host inflammatory response to viruses in the lungs.

## 2. Materials and Methods

### 2.1. Animal Models

C57BL/6 WT, K18hACE2 (hACE, Jackson Laboratories, Bar Harbor, ME, USA), and Cyp KO (gift from Dr. Hector DeLuca, University of Wisconsin, Madison, WI, USA) mice were bred (mice) and housed (mice and hamsters) according to approved IACUC protocols at the Pennsylvania State University (University Park, PA, USA). For the experiments, age- and sex-matched mice were fed: chow diets (D+) (Lab diets #5053, Arden Hills, MN, USA) or purified diets with (D+) and without (D−) vitamin D (Envigo, T.D. 89123, Madison, WI, USA). For some experiments, D+ mice were fed corn oil alone or corn oil with 1,25D. For some experiments, D− mice were fed corn oil alone or corn oil with one of two doses of vitamin D3 (Sigma-Aldrich, C9756, St. Louis, MO, USA). Age- and sex-matched Golden Syrian Hamsters were purchased from Envigo (Indianapolis, IN, USA) and maintained on the chow (D+) or D− diet (Envigo, T.D.120008) and orally fed corn oil or corn oil with vitamin D3. Serum was collected to monitor the vitamin D status of mice and hamsters. 

### 2.2. Serum 25 Hydroxy Vitamin D (25D) Measurements

Serum 25D levels were measured using an ELISA kit and standards as per the manufacturer’s instructions (25-OH D, Eagle Biosciences, Amherst, NH, USA). The limits of detection were 1.6 ng/mL 25D.

### 2.3. SARS-CoV-2 Infection

SARS-CoV-2 USA-WA-1/2020 (Centers for Disease Control and Prevention, BEI Resources, NIAID, NIH: NR-52281) was used for infecting both mice and hamsters. hACE2 mice (*n* = 7–8/group) were anesthetized using isofluorane and infected with 100−1000 TCID_50_ units of SARS-CoV-2 in 50 μL phosphate buffered saline. Hamsters were sedated with ketamine (150 mg/kg), atropine (0.015 mg/kg), and xylazine (7.5 mg/kg) via intraperitoneal injection and intranasally inoculated with 10,000 TCID_50_ units of SARS-CoV-2 in 100 μL Dulbecco’s Modified Eagle Media. Hamsters were given atipamezole (1 mg/kg) subcutaneously to reverse the sedation. Equal numbers of males and females were used for all experiments, and body weights and symptoms were monitored daily until the endpoint criteria were reached or day 14 post-infection.

One series of experiments in hACE2 mice used standard D+ rodent chow (Lab diets #5053, Arden Hills, MN, USA) diets with oral dosing of corn oil or 10 ng/day of 1,25D diluted in 10 μL of corn oil beginning the day before infection and continuing until sacrifice. Additional experiments used mice or hamsters fed D− diets with or without oral vitamin D3 (*n* = 8–16/group). D− hACE2 mice were dosed orally with corn oil (D−), 0.125 μg vitamin D3/day (D+), or 2.5 μg vitamin D3/day (D++) beginning 8 weeks prior to infection and continuing throughout the experiment (*n* = 12–18/group). Hamsters were placed on D− diets with corn oil or 8 μg vitamin D3 (D+) in corn oil/day starting 11 days before infection and continuing throughout the experiment (*n* = 5–6/group).

### 2.4. H1N1 Infection

D+ and D− WT and D+ and D− Cyp KO littermates (*n* = 12–18/group) were fed identical diets with and without vitamin D. Mice were anesthetized with isofluorane to inoculate them intranasally with 10–30 TCID_50_ mouse-adapted A/H1N1/California/04/2009 influenza in 50 μL Gibco’s reduced-serum minimum essential media (ThermoFisher, 31985-070, Waltham, MA, USA) [26]. Body weight and clinical signs were monitored daily until they either reached end-point criteria or at day 14 post-infection.

### 2.5. Biocontainment and Animal Care and Use

All studies with SARS-CoV-2 were conducted in a biosafety level 3 enhanced (BSL3+) laboratory. This facility is approved for BSL3+ respiratory pathogen studies by the U.S. Department of Agriculture and Centers for Disease Control. Studies with pandemic H1N1 influenza were conducted under biosafety level 2 enhanced conditions. All animal studies were conducted in compliance with the Animal Care and Use Committee under protocol numbers: 202001693, 202001516, 202001440, and 202001638. 

### 2.6. RNA Isolation and Quantitative PCR

Tissues were homogenized in TRIzol reagent (Sigma-Aldrich, St. Louis, MO, USA). Total RNA was extracted from the tissues using chloroform-isopropanol precipitation and quantified using NanoDrop (ThermoFisher, Waltham, MA, USA). A total of 1–2 µg RNA was reverse transcribed using AMV Reverse Transcriptase (Promega, Madison, WI, USA). Primers were purchased from Integrated DNA Technologies (IDT, Coralville, IA, USA) and are listed in Appendix A. Gene expression was quantitated using the SYBR green mix (Azura Genomics, Raynham, MA, USA) and StepOne Plus system (Applied Biosystems, Carlsbad, CA, USA). Primers for the SARS-CoV-2 N gene were a commercial kit from IDT (Cat #10006713). Gene expression was calculated using the delta-delta C_T_ method using GAPDH and uninfected control tissues. Gene expression was normalized to day 0 uninfected controls.

### 2.7. Histology

Lung tissues were collected from hamsters and mice and fixed in 10% formalin. These tissues were embedded in paraffin, sectioned, and H & E stained by the PSU Animal Diagnostic Lab Histology lab. Scoring of lung pathology associated with SARS-CoV-2 infection was conducted by veterinary pathologists certified by the American College of Veterinary Pathologists that were blinded to vitamin D3 treatment status. The criteria for the histopathologic evaluation of the tissue sections were conducted as described for H1N1 infected mouse lungs (*n* = 2–5/group) [27,28,29], SARS-CoV-2 infected mouse lung sections (*n* = 4–6/group) [30,31], and hamster lung sections (*n* = 4/group) [32,33,34]. The individual parameters evaluated the extent of pneumonia, damage to alveoli, and lymphocytic infiltration by various specific parameters described in Appendix A.

### 2.8. Statistical Analysis

Results are represented as the mean ± SEM. Statistical analysis was performed using Prism ver. 10 (GraphPad, San Diego, CA, USA) using multiple *t*-tests, one-way ANOVA, two-way ANOVA, mixed-effects analysis with Bonferroni multiple comparisons tests, the Kruskal–Wallis test with Dunn’s multiple comparison, the unpaired *t*-test, and the log-rank survival test as applicable. Data were checked for normal distribution and outliers. *p* < 0.05 was the cutoff to determine significance.

## 3. Results

### 3.1. The Effect of Vitamin D on H1N1 Infection

To evaluate the role of vitamin D deficiency during respiratory virus infection, WT and Cyp KO mice were fed a D+ or D− diet. Regardless of the genotype, serum 25D levels were significantly different between mice fed the D+ or D− diet (Figure 1A). As expected, D+ Cyp KO mice accumulated 25D and had higher levels of 25D than D+ WT mice [25]. Following infection with pandemic H1N1 influenza, respiratory distress was evident in all mice by 7 days post-infection (Figure 1B). D+ WT mice showed the least amount of respiratory distress, and after day 10, D+ WT and D+ Cyp KO mice no longer exhibited symptoms of respiratory distress (Figure 1B). D− WT and D− Cyp KO mice had greater symptoms of respiratory distress that were not completely resolved by day 14 post-infection in the D− Cyp KO mice (Figure 1B). Only 1 of 18 D+ WT mice died following H1N1 infection (Figure 1C). D− WT mice had lower survival compared to D+ WT mice (Figure 1C). Conversely, both Cyp KO groups showed decreased survival; 62% from D+ Cyp KO and 57% from D− Cyp KO mice, respectively. The expression of the influenza M gene at day 4 post-infection was not different in the four groups of influenza-infected mice (data not shown). Lung sections from D− WT and D− Cyp KO mice (d0) but not D+ WT mice showed signs of inflammation even before infection (Figure 1D). The amount of alveolar hemorrhage was significantly more at day 4 post-infection in the D− Cyp KO compared to D+ WT or D− WT (Figure 1D). At day 14 post-infection, the lung sections of D+ Cyp KO, D− WT, and D− Cyp KO mice appeared more severe than D+ WT (Figure 1E). Therefore, vitamin D deficiency and the inability to produce 1,25D increased the susceptibility of mice to H1N1 influenza infection.

### 3.2. Mouse and Hamster Models of SARS-CoV-2 Infection

SARS-CoV-2 does not effectively infect WT mice. The transgenic expression of human (h)ACE-2 was shown to allow SARS-CoV-2 infection in mice [31]. hACE-2 mice were infected with 100, 200, and 315 TCID_50_ SARS-CoV-2, and the mice were evaluated for pre-determined euthanasia endpoints for up to 14 days post-infection. The dose of SARS-CoV-2 that resulted in the sacrifice of 50% of the mice was 200 TCID_50_ (Figure 2A). The mice infected with 100 TCID_50_ failed to reach the euthanasia endpoints, were sacrificed at day 14 post-infection and had only minimal weight loss (Figure 2A). The viral gene copy number for the nucleocapsid (N) protein was measured in the lungs of mice at 100 and 200 TCID_50_. High copy numbers of the N gene were detected as early as day 2 post-infection and remained high until day 6 post infection (Figure 2B). The N gene copy number significantly decreased by day 14 post-infection (Figure 2B). Interestingly, the mice that were infected with 100 TCID_50_ had high amounts of the N gene in the lung even though they showed only mild symptoms of infection, and there were no differences between the N gene expression in the lung between the 100 and 200 TCID_50_ inoculum (Figure 2A,B). Lungs from SARS-CoV-2-infected mice were used to measure mRNA for the 1alpha hydroxylase that produces 1,25D (*Cyp27B1*), the vitamin D receptor (*Vdr*), and the 24 hydroxlase that degrades vitamin D (*Cyp24A1*). There was no change in *Vdr* expression in the lung at day 2 or day 4 post-infection (Figure 2C). Conversely, *Cyp27B1* and *Cyp24A1* expression was higher at day 2 and day 4 post-infection than in the uninfected lung (Figure 2C). *Cyp27B1* expression was significantly lower in the day 4 than the day 2 post-infection lung (Figure 2C). Infection of the hACE2 mice with SARS-CoV-2 induced the expression of two genes that regulate vitamin D metabolism in the lungs.

Hamsters infected with SARS-CoV-2 are reflective of human SARS-CoV-2 infection in that the virus infects the lower respiratory tract, causing similar respiratory sequelae [35]. We previously showed that hamsters infected with 10^5^ TCID_50_ SARS-CoV-2 lost weight shortly after infection, and none of the hamsters died following infection [33]. The weight loss peaked by day 6 post-infection, and the hamsters recovered completely by day 10 post-infection [33].

### 3.3. The Effect of Vitamin D on SARS-CoV-2 Infection

To establish differing vitamin D statuses, hACE2 mice were given a vitamin D deficient chow and were orally dosed with the vehicle (D−), 0.125 μg/day (D+), or 2.5 μg/day (D++) for 8 weeks prior to infection. Serum 25D levels were higher in D+ mice and significantly higher in D++ mice before and at 14 days post-infection (Figure 3A). The D+ dose was inadequate to raise serum 25D levels significantly over the D− values (Figure 3A). The survival of the SARS-CoV-2-infected mice was not affected by the D+ or the D++ treatments (Figure 3B). At day 6 post-infection the amount of N gene expression was the same in D− and D++ mice (Figure 3C). The expression of the *Vdr, Cyp27B1*, *Cyp24A1* was not different in D− and D++ mice at day 6 post-infection (Figure 4A). *Ifnb* and *Ifng* were induced by SARS-CoV-2 infection, while *Ifna* was not (uninfected control set at 1, Figure 4B). Expression of *Ifnb* was significantly lower in the D++ lung as compared to the D− lung at day 6 post-infection (Figure 4B). Mice surviving until day 14 post-infection showed no difference in lung histopathology scores between D− and D+ mice (Figure 3D,E). The D++ lung histopathology scores showed significantly reduced type II hyperplasia, significantly reduced alveolar remodeling, and lower (not significant) total histopathology scores (Figure 3D,E). The final series of experiments tested whether the active form of vitamin D (1,25D) could prevent the lethality of SARS-CoV-2 infection. There was no effect of 1,25D on survival from a lethal dose of SARS-CoV-2 (1,000 TCID_50_) (Figure 3F). There was a trend for faster weight recovery in the surviving 1,25D-treated mice that did not reach significance (Figure 3F). There was an effect of D++ treatment to decrease lung histopathology and a trend towards faster recovery in 1,25D-treated mice infected with SARS-CoV-2.

Serum 25D levels in hamsters fed on the chow diet were not significantly different than serum 25D levels in hamsters fed D− diets for 4 weeks (Figure 5A). Therefore, to control for the diet, hamsters were fed D− diets and then fed orally with the vehicle (D−) or with 8 μg/day of vitamin D3 (D+) beginning 14 days before the SARS-CoV-2 infection and continuing throughout the experiment. Confirming the effectiveness of the dietary intervention, the serum 25D levels were significantly higher in D+ hamsters as compared to D− hamsters before infection on day 0, and this effect was maintained until day 14 post-infection (Figure 5A). The SARS-CoV-2 N gene was detected on day 3 but not at day 6 post-infection in the lungs (Figure 5B), and there was no difference in N gene expression between the D+ versus D− lungs (Figure 5B). Surprisingly, SARS-CoV-2 N gene expression was also detected in the colon tissues of hamsters at both day 3 and day 6 post-infection compared to uninfected control tissues (Figure 5B). Expression of the N gene in the colon was 1000-fold less than in the infected lung (Figure 5). N gene expression was significantly higher in the D− colon compared to baseline values from uninfected tissue controls but not different from uninfected controls in the D+ colon at day 3 post-infection (Figure 5B). The histopathology of the lungs showed significantly more damage at day 6 than day 3 post-infection (Figure 5C,D). There was no effect of vitamin D on the weight loss or histopathology scores following SARS-CoV-2 infection of the hamsters (Figure 5D,E).

## 4. Discussion

Vitamin D deficiency resulted in lung inflammation in the absence of infection. Infected D− mice had more severe lung inflammation and respiratory symptoms than D+ or D++ mice when infected with either H1N1 influenza or SARS-CoV-2. The data point to shared effects of vitamin D to control inflammation in the lung following influenza or coronavirus infection. D− mice had significantly more inflammation than D+ mice following H1N1 influenza infection (Figure 1). High-dose vitamin D3 treatment (D++) resulted in some protection of mice from SARS-CoV-2 (Figure 3). In addition, 1,25D treatment showed a trend towards faster recovery of surviving mice from SARS-CoV-2 (Figure 3). Others have shown that 1,25D-treated mice had reduced lung inflammation [20,21], and treating D+ mice with 25D had a small protective effect on weight loss and lethality following H1N1 infection [22]. A recent clinical trial that used 25D in humans showed reduced mortality in hospitalized patients with COVID-19 [36,37]. However, it is unclear whether 25D treatment would be effective in mouse or hamster models of SARS-CoV-2. This is the first study that investigated the effects of vitamin D in animal models of SARS-CoV-2. The data suggest that vitamin D and 1,25D may be effective to protect the lung from SARS-CoV-2. SARS-CoV-2 infection induced *Cyp27B1* and *Cyp24A1* in the lung of mice suggesting a role for vitamin D metabolites in the lung response to SARS-CoV-2 infection (Figure 2). There are likely shared and unique mechanisms by which vitamin D regulates the host response to influenza versus SARS-CoV-2. A better understanding of the mechanisms by which vitamin D regulates the anti-viral response in the lung to both influenza and coronaviruses is needed to inform clinical studies.

Cyp27B1 KO mice cannot produce 1,25D, which induces Cyp24A1 and degrades 25D and 1,25D. Feeding Cyp27B1 KO mice D+ diets results in the accumulation of 25D (Figure 1A, [25,38]). 25D is a low-affinity ligand for the VDR, and it was shown that high amounts of 25D can replace the need for 1,25D for the regulation of calcium homeostasis and osteomalacia [39]. Previous experiments showed that D+ Cyp KO and D+ WT mice cleared a bacterial infection in the gut with similar kinetics [25]. Conversely, D+ Cyp KO mice had higher lethality and more severe inflammation than D+ WT mice when infected with H1N1 influenza (Figure 1). The effects of Cyp27B1 expression on host resistance to a bacteria could be different than the effects on host resistance to a virus. It would be interesting to determine the effect of the Cyp27B1 deletion on host resistance to other respiratory viruses including SARS-CoV-2. Conversely, the differential effect of Cyp27B1 could be due to the location of the infection in the gut versus the lung. Regardless, it seems that the ability of the host to produce Cyp27B1 is important for the mice to survive a H1N1 lung infection.

The dietary interventions to generate D−, D+, and D++ hACE2 mice resulted in D++, but not D+, mice having higher serum 25D than D− mice (Figure 4). Interestingly, the hamster studies suggest that the commercially available chow may not be adequate to raise serum 25D levels (Figure 5A). The data suggest that the adequacy of vitamin D should be considered in evaluating studies that infect chow-fed hamsters with SARS-CoV-2. At day 6 post-SARS-CoV-2 infection, the D++ hACE2 mice had less inflammation and lower IFN-β in the lung than the D− mice (Figure 3). The results are consistent with the anti-inflammatory effects of vitamin D. Suppression of type-1 inflammatory cytokines by vitamin D underlies the effects of vitamin D and 1,25D to suppress immune-mediated diseases [40,41,42]. The benefits of 25D from influenza infection were associated with a reduction in IFN-γ in the lung [22]. Recently, Chauss et.al showed that 1,25D promotes anti-inflammatory responses by switching off IFN-γ production from Th1 cells and upregulating IL-10 [43]. IFN-γ and IFN-β production is essential for effective viral clearance; viruses have mechanisms to evade the IFN responses, and severe COVID-19 is associated with dysregulation of IFN responses [44,45,46]. Down-regulation of IFN-β by vitamin D is associated with protection from inflammation in the lung following a virus infection with either influenza or SARS-CoV-2.

Importantly, there was no effect of vitamin D on the SARS-CoV-2 N gene or H1N1 M gene expression in the lungs of mice or hamsters. This indicates that vitamin D did not reduce or inhibit viral replication in the lung. We found SARS-CoV-2 N gene expression in the hamster colons. Interestingly, D− colons had relatively more SARS-CoV-2 N gene expression than D+ colons. The implications of having SARS-CoV-2 in the colon but not the lung would need to be determined, and it would be important to quantitate live virus in the tissues. Unfortunately, we did not save colons from our SARS-CoV-2 mouse studies. Vitamin D is shown to be a strong inducer of cathelicidin LL-37 in human cells [47]. There is some evidence that LL-37 can directly kill some viruses including influenza viruses [13,14,15,16,17]. Treating mice with a high dose (500 μg/day) of human LL-37 peptide protected from lethal influenza infection and significantly reduced viral titers at day 3 post-H1N1 infection in the lung [48]. LL-37 inhibited binding of the SARS-CoV-2 spike protein containing pseudo viruses both in vivo and in vitro blocking entry via ACE2 [49]. There were no effects of vitamin D in vivo on the expression of viral genes for SARS-CoV-2 or H1N1 influenza in the lung. The cathelicidin peptides found in mice are not the same as the LL-37 in humans, and the mouse cathelicidin is not regulated by vitamin D [50]. The lack of a vitamin D effect on SARS-CoV-2 was shown in mice and hamster lungs. It is unclear whether the hamster cathelicidin gene has vitamin D response elements. Furthermore, no studies have been conducted to test the effect of vitamin D on SARS-CoV-2 in vitro. Therefore, an effect of vitamin D through the induction of anti-bacterial peptides, such as LL-37, that reduces viral titers cannot be ruled out.

The data from mouse H1N1 and mouse and hamster SARS-CoV-2 infection point towards a protective role of vitamin D against acute viral infection in the lung. The data are in line with several meta-analyses and observational studies suggesting a beneficial effect of vitamin D in the lung [51,52,53,54]. Vitamin D supplementation was reported to have a moderately protective effect during acute respiratory tract infections, such as influenza and COVID-19; however, this outcome was affected by a number of variables such as dose frequency, season of supplementation, and pre-existing conditions [51,52,53,54]. Patients with existing vitamin D deficiency, especially patients >80 years of age, are at a higher risk of being infected with SARS-CoV-2 and developing severe disease [51,52,53,54]. High-dose vitamin D supplementation with either calcifediol or cholecalciferol was shown to reduce overall mortality and severe outcomes such as intensive care admission [55,56,57,58,59]. Vitamin D deficiency is associated with acute respiratory distress syndrome (ARDS), and several clinical trials are underway to investigate whether vitamin D supplementation can reduce the development of ARDS [60]. However, a recent randomized controlled study (reported in medRxiv, [61]) that identified vitamin D deficiency and then treated it showed that vitamin D supplementation did not reduce the risk of acute respiratory infection or the risk of COVID-19 infection. With the limitations of performing a clinical study for a nutrient such as vitamin D, our study in animals suggests that vitamin D does control the host response to H1N1 and SARS-CoV-2 infection in the lung. The data point to an effect of vitamin D to control the cytokine response and resolve inflammation in the lung following a viral infection. Understanding the mechanisms and timing of the vitamin D effects in animal models would inform future clinical trials.

There are several limitations of the current animal studies. There were technical limitations due to the cost, training, and restrictions needed to use the BSL3 facility for SARS-CoV-2 infections safely. Experiments in the BSL-3 facility had small sample sizes. In addition, viruses including H1N1 and SARS-CoV-2 do not naturally infect mice, and so the viruses used were passaged (H1N1), or a transgenic mouse was needed to allow infection (hACE) with SARS-CoV-2. In addition, vitamin D metabolism is not identical in animals and humans. To our surprise, hamsters fed on standard chow diets replete in vitamin D were found to have low serum 25D levels. It is possible that vitamin D metabolism in hamsters is different than in the mouse and humans, which might impact our results. Mechanistic studies in humans are difficult, so it is important to use animals to determine pathways and processes that are regulated by vitamin D. The data in humans and animals do support a role of vitamin D to control the inflammatory responses in the lung providing protection following a viral infection with H1N1 or SARS-CoV-2.

## 5. Conclusions

Together, the data support an important role for vitamin D and Cyp27B1 in the regulation of the host response to H1N1 and SARS-CoV-2 viruses. The role of vitamin D includes the restraining of the IFN response shortly after infection. Vitamin D-deficient hosts had pre-existing inflammation in the lungs that contributed to susceptibility to viral infection. Future experiments should continue to determine the mechanisms by which vitamin D regulates the anti-viral response in the lungs and whether there are differences in the effect of vitamin D on host resistance to H1N1 influenza and SARS-CoV-2.

## Figures and Tables

**Figure 1 nutrients-14-03061-f001:**
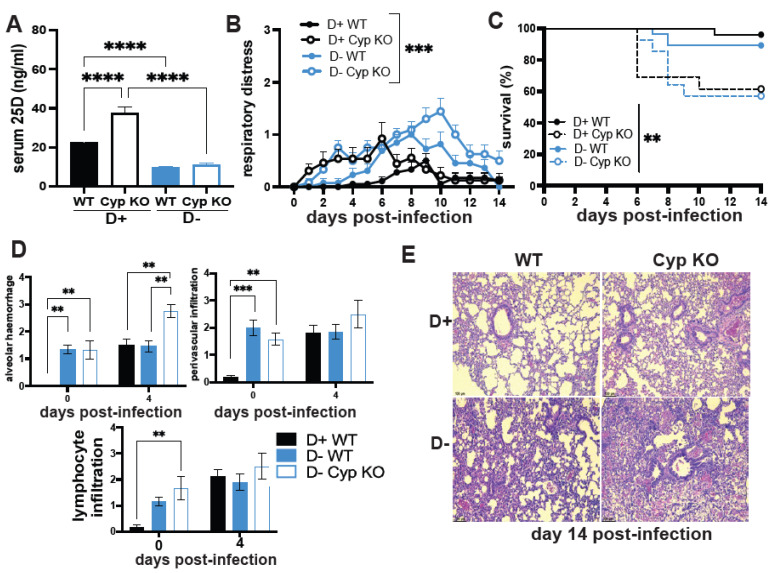
Vitamin D deficiency and Cyp27B1 KO increases susceptibility to H1N1 infection. D+ and D− WT and D+ and D− Cyp KO littermates were infected with H1N1 influenza (*n* = 12–18 per group). (**A**) Serum was collected at day 14 post-infection to measure 25D. Mice were monitored for (**B**) respiratory distress symptoms and (**C**) survival of D+ WT (*n* = 18), D+ Cyp KO (*n* = 13), D− WT (*n* = 13), and D− Cyp KO (*n* = 12) mice until day 14 post-infection. Lung tissues were collected for histology from uninfected and day 4 post-infected mice for histology. H & E-stained sections were scored for (**D**) lung alveolar hemorrhage, perivascular infiltration, and lymphocyte infiltration in D+ WT (*n* = 4) and D− WT (*n* = 4–5) and D− Cyp KO (*n* = 2–3) mice at day 0 and day 4 post-infection. (**E**) Representative histology of the lung of D+ WT (score = 3), D+ Cyp KO (score = 4), D− WT (score = 5.5), and D− Cyp KO (score = 6) at day 14 post-infection. Sections from D− WT, D+ Cyp KO, and D− Cyp KO show increased lymphocyte infiltration, alveolar hemorrhage and constricted bronchiolar spaces compared to D+ WT. Values are the mean ± SEM. Statistical significance was assessed using one-way ANOVA with Bonferroni multiple comparison test for (**A**) two-way ANOVA with Bonferroni multiple comparison test for (**B**,**D**) and log rank (Mantel–Cox) survival analysis for (**C**). ** *p* < 0.01, *** *p* < 0.001, and **** *p* < 0.0001.

**Figure 2 nutrients-14-03061-f002:**
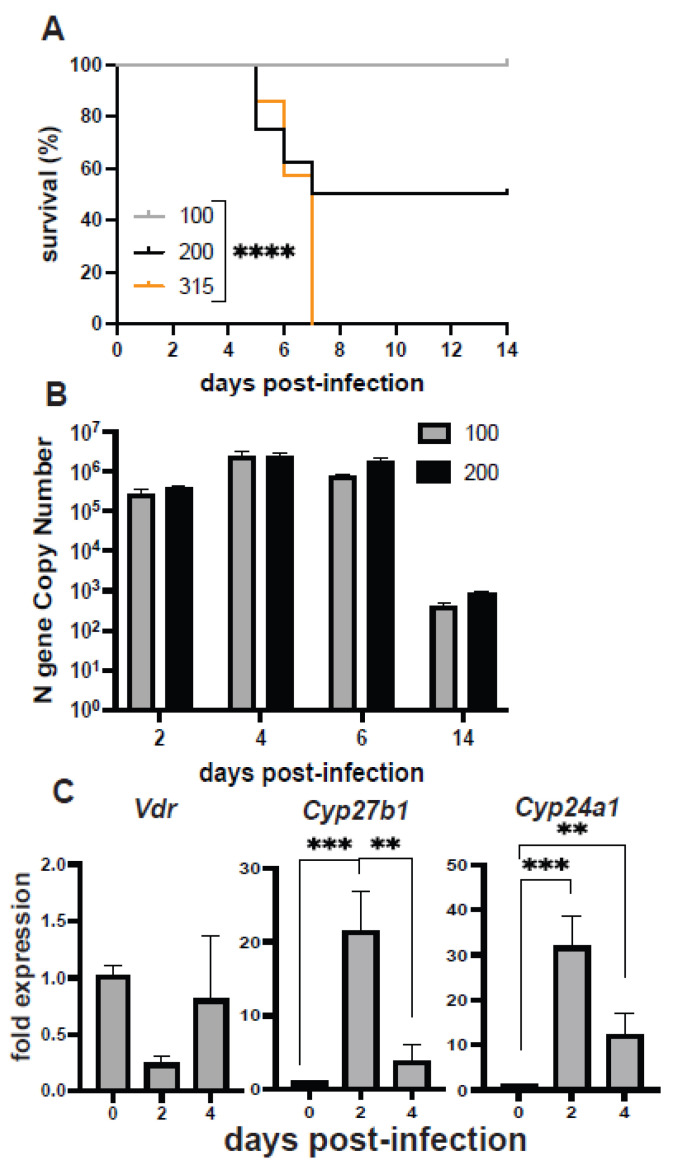
SARS-CoV-2 infection. K18-hACE2 mice were intranasally inoculated with 100 (*n* = 8), 200 (*n* = 8), and 315 (*n* = 7) TCID_50_ of SARS-CoV-2 virus. Mice were sacrificed when they met pre-determined endpoints (worsening conjunctivitis, lethargy, labored breathing, and/or dehydration) or at day 14 post-infection. (**A**) Mice were monitored for survival until day 14 post-infection. Mice were euthanized, and lung tissue was collected for (**B**) SARS-CoV-2 N gene expression at days 2–14 post-infection, (**C**) *Vdr, Cyp24A1*, and *Cyp27B1* mRNA gene expression (*n* = 4/group/timepoint) at day 0, day 2, and day 4 post-infection in the lung. Samples were normalized to uninfected control tissue. Values are the mean ± SEM. Statistical significance was assessed using log-rank test for trend (**A**), two-way ANOVA model for main effects only for (**B**), and one-way ANOVA with Bonferroni multiple comparison test on log-transformed expression values for (**C**). ** *p* < 0.01, *** *p* < 0.001, and **** *p*< 0.0001.

**Figure 3 nutrients-14-03061-f003:**
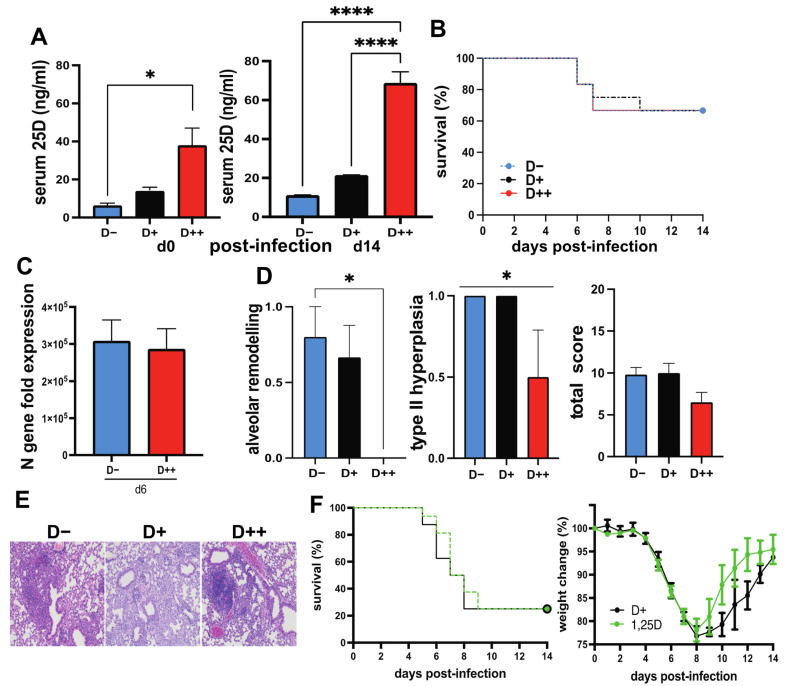
High dose vitamin D reduces lung inflammation following SARS-CoV-2 infection. hACE2 mice were fed vitamin D deficient (D−, *n* = 18), vitamin D sufficient (D+, *n* = 12), or vitamin D supplemented (D++, *n* = 18) diet and infected with 200 TCID_50_ SARS-CoV-2. (**A**) Serum 25D was measured before and at day 14 post-infection, and (**B**) the survival of D−, D+, and D++ mice. (**C**) SARS-CoV-2 N gene expression in the lungs (*n* = 6 mice/group) at day 6 post-infection. Gene expression relative to uninfected D+ controls. (**D**) Alveolar remodeling, type II pneumocyte hyperplasia, and total histology score in D−, D+, and D++ mice (*n* = 4–6 mice/group) at day 14 post-infection. *(***E**) Representative histology images for D− (score = 9), D+ (score = 11), and D++ (score = 5). D+ hACE2 (*n* = 8) mice at day 14 post-infection. 1,25(OH)_2_D (1,25D)-treated D+ hACE 2 (*n* = 16) mice were infected with 1000 TCID_50_ SARS-CoV-2. (**F**) Survival and body weight change over the course of infection. Values are the mean ± SEM. Statistical significance was assessed using one-way ANOVA with Bonferroni multiple comparison test for (**A**,**D**), log rank (Mantel–Cox) test for each of the groups for (**B**,**F**), unpaired *t*-test on log-transformed expression values for (**C**), and two-way ANOVA with Bonferroni multiple comparison test for (**F**). * *p* < 0.05 and **** *p* < 0.0001.

**Figure 4 nutrients-14-03061-f004:**
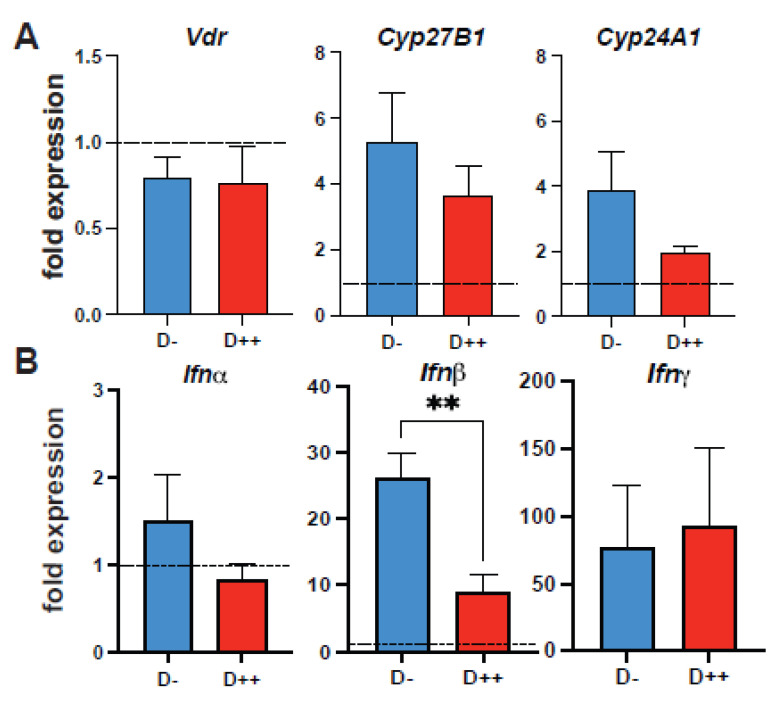
D++ mice have lower *Ifnb* at day 6 post-SARS-CoV-2 infection in the lung. Lung mRNA from day 6 SARS-CoV-2 infected D− (*n* = 5) and D++ (*n* = 5) mice. (**A**) *Vdr*, *Cyp27B1*, *Cyp24A1*, and (**B**) *Ifna*, *Ifnb*, and *Ifng* relative to *Gapdh* and uninfected controls set at 1 (dashed line). Values are the mean + SEM. Statistical significance was assessed using unpaired *t*-test on log-transformed values. ** *p* < 0.01.

**Figure 5 nutrients-14-03061-f005:**
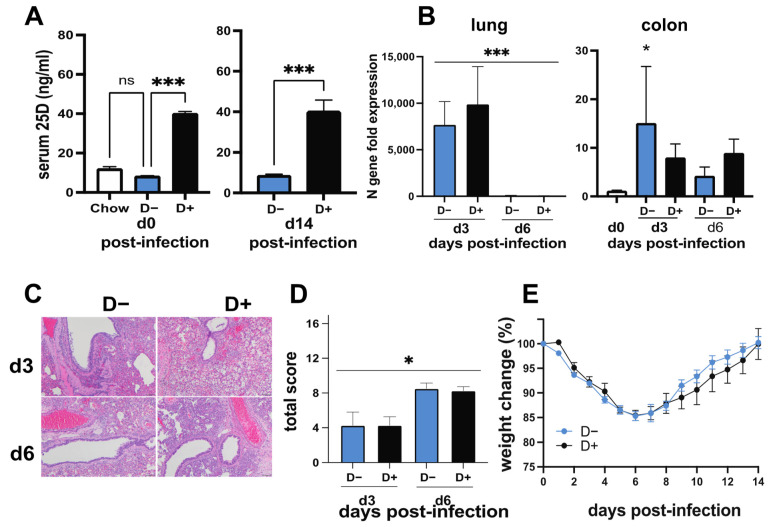
SARS-CoV-2 infection of hamsters. Hamsters were fed D− diets with corn oil (D−) or 8 μg D3/day dissolved in corn oil (D+) and then infected with SARS-CoV-2 virus. (**A**) Serum levels of 25D in chow-fed, D+, and D− fed hamsters before (*n* = 2–10 hamsters/group) or from D− and D+ hamsters 14 days after (*n* = 5–6 hamsters/group) SARS-CoV-2 infection. Hamsters were euthanized at day 3 and day 6 post-infection, and tissues were collected for gene expression as well as histology. (**B**) SARS-CoV-2 N gene expression in the lung and colon relative to uninfected control (*n* = 4 hamsters per group and timepoint). (**C**) Representative histopathology sections of the lungs at day 3 (D− score = 6, D+ score = 4) and day 6 (D− score = 10, D+ score = 9) post-infection showed increased lymphocyte infiltration and lesions in D− and D+ hamsters at day 6 post-infection and (**D**) total histopathology scores (*n* = 4 hamsters/group) were determined based on the scoring criteria outlined in Appendix A. (**E**) Change in body weight following infection with SARS-CoV-2 (*n*= 5–6 hamsters/group). Values are the mean ± SEM. Statistical significance was assessed using Kruskal–Wallis test with Dunn’s multiple comparison (day 0) and unpaired *t*-test (day 14) for (**A**), one-way ANOVA with Bonferroni multiple comparison test on log-transformed expression values for (**B**), one-way ANOVA for (**D**), and two-way ANOVA with Bonferroni multiple comparison test for (**E**). * *p* < 0.05 and *** *p* < 0.001.

## Data Availability

Not applicable.

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
