# Peer review of "Vitamin D and the Ability to Produce 1,25(OH)2D Are Critical for Protection from Viral Infection of the Lungs"

_nutrients, 2022, doi:10.3390/nu14153061_

Round 1

Reviewer 1 Report

As the authors note, cause and effect are extremely difficult to determine in human studies. Accordingly, the authors we sought to evaluate the effects of vitamin D on the lung anti-viral response in animal models. This is especially important given the potential for reverse causality in human virus study.  For example, a disease like C-19 may actually lower Vit-D levels rather than the opposite direction. Also, animal studies are not prone to confounding factors such as corticosteroid use (see PMID:35440302). Overall, this is a rigorous and well-written manuscript.  My comments are mostly minor.

1)   While some sample size information is provided in the figure legends, this also should be explicitly stated in the methods section.

2)  A few sentences summarizing the limitations of the study should be provided in the discussion section.

3) In  the legends of Figures 1 and 3, “Log-rank (Mankel-Cox)” should be “Log-rank (Mantel-Cox)”.

Author Response

1)   While some sample size information is provided in the figure legends, this also should be explicitly stated in the methods section. RESPONSE:  We have gone through the methods to add the sample size.

2)  A few sentences summarizing the limitations of the study should be provided in the discussion section.RESPONSE:  We have added a paragraph that discusses the limitations of the study.

3) In  the legends of Figures 1 and 3, “Log-rank (Mankel-Cox)” should be “Log-rank (Mantel-Cox)”. RESPONSE: We have corrected our error, thank you for pointing it out.

Reviewer 2 Report

Dear Authors,

I read paper titled:

Vitamin D and the ability to produce 1,25 (OH) 2D are critical for protection from viral infection of the lungs.

The article is extremely interesting and brings a series of information to the literature. I recommend the following:

1. In Figure 1 - please enter a more detailed description. It is easier for readers to understand. Also, for Fig1 (E) please describe in more detail

2. Such a recommendation for Figure 2 and 5

3. The discussion subchapter is too short. I recommend detailing and exemplifying other studies in the field, comparing other results with yours. I also recommend a more comprehensive description of the possible clinical impact.

4. I recommend, in the Discussion Chapter, to introduce examples of clinical studies and possible influences on the clinical prognosis.

BR

Author Response

  1. In Figure 1 - please enter a more detailed description. It is easier for readers to understand. Also, for Fig1 (E) please describe in more detail. RESPONSE: We have added a more detailed description to the figure legend.
  2. Such a recommendation for Figure 2 and 5. RESPONSE: We have added a more detailed description to the figure legend.
  3. The discussion subchapter is too short. I recommend detailing and exemplifying other studies in the field, comparing other results with yours. I also recommend a more comprehensive description of the possible clinical impact. RESPONSE: We have added additional discussion that we hope does a better job of discussing the results in the field.
  4. I recommend, in the Discussion Chapter, to introduce examples of clinical studies and possible influences on the clinical prognosis. RESPONSE: We have added discussion about the implications for clinical studies.